# The Effect of PEGDE Concentration and Temperature on Physicochemical and Biological Properties of Chitosan

**DOI:** 10.3390/polym11111830

**Published:** 2019-11-07

**Authors:** Martha Gabriela Chuc-Gamboa, Rossana Faride Vargas-Coronado, José Manuel Cervantes-Uc, Juan Valerio Cauich-Rodríguez, Diana María Escobar-García, Amaury Pozos-Guillén, Julio San Román del Barrio

**Affiliations:** 1Unidad de Materiales, Centro de Investigación Científica de Yucatán, México. Calle 43 No. 130 x 32 y 34, Colonia Chuburná de Hidalgo, C.P. 97205 Mérida, Yucatán, Mexico; martha.chuc@correo.uady.mx (M.G.C.-G.); ross@cicy.mx (R.F.V.-C.); manceruc@cicy.mx (J.M.C.-U.); 2Laboratorio de Ciencias Básicas, Facultad de Estomatología, Universidad Autónoma de San Luis Potosí, México. Ave. Dr. Manuel Nava No. 2, Zona Universitaria, C.P. 78290 San Luis, S.L.P., Mexico; diana.escobar@uaslp.mx (D.M.E.-G.); apozos@uaslp.mx (A.P.-G.); 3Instituto de Ciencia y Tecnología de Polímeros. España. Calle Juan de la Cierva, 3, C.P 28006 Madrid, Spain; jsroman@ictp.csic.es

**Keywords:** chitosan, crosslinked, PEGDE, glutaraldehyde, drying temperature, biocompatibility, scaffolds, osteoblasts

## Abstract

Chitosan (CHT) is a polysaccharide with multiple claimed properties and outstanding biocompatibility, generally attributed to the presence of protonable amino groups rendering a cationic natural polymer. However, the effect of changes in CHT structure due to hydration is not considered in its performance. This study compares the effects on biocompatibility after drying at 25 °C and 150 °C scaffolds of chitosan, polyethylene glycol diglycidyl ether (PEGDE) crosslinked CHT (low, medium and high concentration) and glutaraldehyde (GA) crosslinked CHT. PEGDE crosslinked CHT showed a reduction in free amino groups and the amide I/II ratio, which exhaustive drying reduced further. In X-ray diffraction (DRX) analysis, PEGDE crosslinked CHT showed multiple peaks, whereas the crystallinity percentage was reduced with an increase in PEGDE concentration and thermal treatments at 150 °C. In a direct contact cell assay, high osteoblast viability was achieved at low and medium PEDGE concentrations, which was improved when the crosslinked scaffolds were thermally treated at 150 °C. This was attributed to its partial hydrophilicity, low crystallinity and low surface roughness; this in spite of the small reduction in the amount of free amino groups on the surface induced during drying at 150 °C. Furthermore, PEGDE crosslinked CHT scaffolds showed strong vinculin and integrin 1β expression, which render them suitable for bone contact applications.

## 1. Introduction

Tissue repair makes use of different strategies, including scaffold-free, cell-free and traditional tissue engineering, where cells are seeded on biodegradable scaffolds in the presence of bioactive molecules while being stimulated in a bioreactor. Therefore, scaffolds are an important element for tissue repair, because they not only provide mechanical support, but also facilitate cell proliferation and differentiation through their porous structure. The materials used to produce scaffolds are natural or synthetic polymers, ceramics and even biodegradable metals [1]. 

Among them, natural polymers have caught the attention of several researchers due to their diverse and sometimes unexpected properties, such as reactivity, strength and toughness, when blended with other polymers [2,3].

Chitosan (CHT) is the N-deacetylated derivative of chitin with a typical DA of less than 0.35. It is thus a copolymer composed of glucosamine and N-acetylglucosamine. The physical properties of CS depend on several parameters such as the molecular weight (from approximately 10,000 to 1 million Dalton), DD (in the range of 50–95%), sequence of the amino and the acetamido groups and the purity of the product [4].

CHT has multiple claimed properties, such as biocompatibility [5,6,7], anti-inflammatory [8], antimicrobial [9,10], hypocholesterolemic [11], immunostimulant [12], antitumor activity [13], antioxidant and anticancer properties [14]. Because of its no-toxicity, non-allergenicity, compatibility with other materials [15] and biocompatibility, chitosan is an ideal candidate for both the food and pharmaceutical industries, including dental applications and tissue engineering [15,16,17,18,19]. In addition, the presence of amino groups along the polymer chain allows chitosan to form complexes with anionic molecules as lipids, DNA, proteins and negatively-charged, synthetic polymers, which expand its range of application in drug delivery and gene therapy [20].

The content of amino groups, degree of deacetylation and molecular weight are the main reasons for the diverse physicochemical properties of chitosan [6]. In terms of biodegradation some factors that can influence this behavior include the distribution of acetamide groups [21], degree of crystallinity [22], type and concentration of crosslinking agents [23,24,25] and degradation conditions (pH and temperature, and the presence of enzymes) [26]. 

In addition, it is known that CHT is a semicrystalline material with a polymorphic crystalline structure that can be transformed from its hydrated forms to an anhydrous or “annealed” form. This quality of the CHT was reported for the first time by Sakurai [27]. 

Surface properties are determinants of CHT cellular response [28,29] but it remains unknown whether changes in the hydration degree will affect CHT biocompatibility.

In this study, we argued that not only the presence of amino groups on CHT are responsible for its biocompatibility, but also that the existence of other factors, such as changes in crystallinity and surface roughness, played a significant role in its cellular response. Therefore, we proposed the use of polyethylene glycol diglycidyl ether (PEGDE) as a crosslinking agent in order to improve CHT mechanical properties, and to assess their biocompatibility with osteoblasts due to amino groups involvement in the crosslinking reaction. Furthermore, we assessed the effect of a high temperature drying treatment on PEGDE crosslinked CHT to fully appreciate if these structural changes will jeopardize CHT biocompatibility. 

## 2. Materials and Methods 

### 2.1. Materials 

Chitosan (CHT) with viscosity molecular weight of 223,332 gmol^−1^, (determined experimentally [24]) and a degree of deacetylation of 84.1% (Batch number: STBF4197V provided by the supplier), polyethylene glycol diglycidyl ether (Mn 500) and glutaraldehyde (GA) solution Grade II, 25% in H_2_O were purchased from Sigma-Aldrich (Saint Louis, MO, USA ), while J.T. Baker provided acetic acid (Xalostoc, México). MTS CellTiter 96^®^ Aqueous Non-Radioactive Cell Proliferation obtained from Promega (Madison, WI, USA) Dulbecco’s modified Eagle’s medium (DMEM) from Biowest (Riverside, MO, USA), human osteoblast hFOB1.19 from ATCC^®^ CRL-11372™ (Manassas, VA, USA). Integrin β1 (A-4): sc-374429, vinculin (7F9) sc-73614 and antibody (normal mouse IgG: sc-3891 SCBT) were purchased from Santa Cruz Biotechnology (Dallas, TX, USA).

### 2.2. Preparation of Chitosan-PEGDE and Chitosan-GA Films

Chitosan (200 mg/30 mL) was dissolved in 0.4 M acetic acid. This solution was stirred in a magnetic stirrer plate at 100 rpm at 25 °C, for one hour until a clear solution was obtained. After this, 25 mL of the solution was poured into plastic Petri dishes, allowing them to dry at 25 °C for at least 24 h. The resulting films were neutralized with sodium hydroxide solution (5 wt.%), washed with distilled water and dried at 25 °C. Crosslinked chitosan was obtained in a similar manner, but by adding either 0.3744 mM (150 μL) of glutaraldehyde or 0.114 mM (50 μL), 0.228 mM (100 μL) and 0.342 mM (150 μL) of PEGDE. This will be referred as PEGDE1, PEGDE2 and PEGDE3, respectively. Stirring continued for 5 h until complete homogenization of the mixture and then the solution was poured into plastic Petri dishes and dried at 25 °C. Films obtained after solvent casting were neutralized with 5 wt.% NaOH and rinsed with distilled water. Films of uncrosslinked and crosslinked chitosan were also thermally treated at 150 °C during 24 h on vacuum conditions. This temperature was selected based on CHT reported *T*g, which depended on water content and heat treatment, [30,31,32] that drying occurs generally after 110 °C, and that the degradation temperature of CHT is above 300 °C [32].

### 2.3. Characterization of GA-crosslinked and PEGDE Crosslinked CHT Films

#### 2.3.1. Fourier-transform-infrared Spectroscopy (FTIR) 

Fourier transform infrared (FTIR) spectra of the films were obtained by using a Thermo Scientific Nicolet 8700 FT-IR spectrometer (Madison, WI, USA) with Zinc Selenide attenuated total reflectance (ATR) accessory. The spectra were acquired in the 4000 and 650 cm^−1^ spectral range averaging 100 scans with a resolution of 4 cm^−1^.

#### 2.3.2. Free Amine Groups Determination

Crosslinked chitosan (0.125 g) was placed in 25 mL of 0.1 M HCl aqueous solution and left for 20 h to allow amine group protonation. Subsequently, the solution was titrated with a 0.1 M NaOH solution and the pH measured with a pH-Ion 510 pH meter (OAKTON Instruments, Vernon Hills, IL, USA). The same procedure was repeated with pure chitosan films. The percentage of free amino groups in the sample was calculated with the following Equations (1) and (2) [33].
(1)$${\mathrm{NH}}_{2}\%=\left[\frac{\left(C_{1}V_{1}-C_{2}V_{2}\right)\times 0.016\times 100}{w\left(100-{\%H}_{2}O\right)}\right]$$
(2)$${\mathrm{NH}}_{2}\%\;\mathrm{free}=\left[\frac{{\mathrm{NH}}_{2}\%}{9.94\%}\right]\times 10$$
where *C*_1_ is the concentration of HCl, *C*_2_ is the concentration of NaOH, *V*_1_ is the volume of the HCl solution, *V*_2_ is the volume of the NaOH solution added during titration, 0.016 is the NH_2_ content in g/mL of 1M HCl and *w* is the mass of the sample (g). The percentage of theoretical NH_2_ content in chitosan was considered as 9.94% [34]. The water content (*w*) was precisely calculated by the mass loss of the sample before and after drying at 150 °C using the Thermogravimetric Analysis (TGA) thermogram.

#### 2.3.3. Thermogravimetric Analysis (TGA) 

Thermogravimetric analysis was conducted in a TGA 8000™ from Perkin-Elmer (Waltham, MA, USA) The analysis was performed on 5–10 mg of the sample after heating from 50 to 700 °C at 10 °C/min under continuous flow of dry nitrogen. 

#### 2.3.4. X-Ray Diffraction (XRD) 

X-ray diffraction patterns of films were recorded by using a Bruker D-8 Advance X-ray diffractometer (Karlsruhe, Germany) Samples were analyzed between 2θ angles of 5° and 60°. The voltage, the current, and the time per step were 40 mV, 55 mA and 6 s, respectively. The crystallinity index (Cr*I*_020_) was calculated using the following Equation (3) [35]
(3)$$\mathrm{Cr}I_{020}=\left[\left(I_{020}-I_{\mathrm{am}}\right)/\left(I_{020}\right)\right]\times 100$$
where *I*_020_ is the maximum intensity below 13° and *I*_am_, the intensity of amorphous diffraction at 16°.

#### 2.3.5. Scanning Electron Microscopy (SEM) 

The samples were coated with gold in a Denton Desk II Sputter Coater (Moorestown, NJ, USA) (50 s, 40 mA). The morphology of film surface was examined using a JEOL, JMS 6360LV (Akishima Tokyo, Japan) with acceleration voltage of 20 kV. Energy-dispersive X-ray spectroscopy (EDX) (Oxford Instruments, INCA X-Sight 7582, High Wycombe, UK) coupled with the microscope, was used to obtained elemental composition. 

#### 2.3.6. X-ray Photoelectron Spectroscopy (XPS)

Spectra were obtained on a Thermo Scientific K-Alpha X-ray Photoelectron Spectrometer (Waltham, MA, USA), using AlEs cathode, without erosion. Survey spectra were acquired over a binding energy range of 0 to 1100 eV, using a pass energy of 50 eV. High-resolution spectra for the C1s, O1s, N1s regions were also obtained. Elemental composition (at.%) was calculated from the integrated intensities of the XPS peaks, which considered the atomic sensitivity factors of the instrument data system. 

#### 2.3.7. Atomic Force Microscopy (AFM)

Surface morphology was obtained with a Bruker INNOVA AFM scanning probe microscope (Santa Barbara, CA, USA), with a commercial silicone tip (RTESP nanoprobe Bruker) at a resonance frequency of 300 kHz, and a spring constant of 40 N/m and 8 nm tip radius. Additionally, a statistical analysis was employed to obtain the roughness of the samples (Ra): First, the scanned area of 100 μm × 100 μm was sectioned in four sub-areas of 50 μm × 50 μm at the scanning frequency of 0.5 Hz; then, roughness was calculated for each subarea using the Nanoscope Analysis software. The statistical average and the standard deviation of the roughness of each type of sample was reported considering 12 measurements.

#### 2.3.8. Contact Angles 

A ramé-hart model 250 goniometer/tensiometer with DROPimage Advanced v2.8 (Succasunna, NJ, USA) was used to measure the contact angle, at room temperature (25 °C), on the surface of the chitosan films. A 10 mm × 60 mm film was placed on a movable sample stage and leveled horizontally. Then, using a microsyringe, 5 μL of either distilled water or Dulbecco’s Modified Eagle’s Medium (DMEM) was deposited on the surface of the film. Ten replicates per sample were averaged. The image of the water or DMEM droplet was captured within 10 s of delivery.

### 2.4. Biocompatibility Studies

#### 2.4.1. Cell Viability and Proliferation Studies 

Cell viability studies were done using CellTiter 96^®^ AQueous Non-Radioactive Cell Proliferation Assay, following manufacturer instructions. Briefly for 34 samples, 648 μL of MTS[3-(4,5-dimethylthiazol-2-yl)-5-(3-carboxymethoxyphenyl)-2-(4-sulfophenyl)-2H-tetrazolium] was mixed with 32 μL PMS (phenazine methosulfate) after thawing. The entire procedure was performed at 25 °C and protected from light. The MTS solutions was used immediately after preparation.

For the direct contact assays were used samples of 10 mg of weight. Scaffolds were sterilized by UV for 15 min. Osteoblasts, obtained from human osteoblast ATCC hFOB1.19, were seeded at 2 × 10^3^ cells/scaffolds in a 96-well plate. The seeded cells were incubated at 37 °C, 5% CO_2_ and 95% humidity for 48 h. 

Extracts were obtained according to the practical guide for the preparation of samples and reference materials, ISO 10993-12 [33]. 25 mg of chitosan scaffolds were UV sterilized during 15 min per side and placed in contact with 10 mL of DMEM (2.5 mg/mL) for 24 h at 24 °C. After this, the solutions were centrifuged and filtered with a Spritzer TPP^®^ syringe filter of 0.22 μm. For an indirect contact test, extracts replaced the culture medium in a 96 well plate with 2 × 10^3^ cells/well. The seeded cells were incubated at 37 °C, 5% CO_2_ and 95% humidity for 48 h.

For both techniques (direct and indirect) a positive control was included which comprised cells with only culture medium (C+), with a negative control (C−), with hydrogen peroxide and with the target with MTS solution, as well.

After 48 h, 20 μL of MTS CellTiter 96^®^ was added to each well. This was incubated for 3 h and the absorbance was measured in a multi-well spectrophotometer (Thermo Scientific™) at 490 nm.

#### 2.4.2. Integrin and Vinculin Expression Analysis 

Cytoskeletal alterations were measured by the expression of integrin and vinculin adhesion proteins. To carry out these tests, extracts obtained from chitosan scaffolds crosslinked with PEGDE were used. A 24-well plate with 120 µL of each sample was seeded with 20 x 10^3^ cells/mL (osteoblast ATCC hFOB1.19), and incubated at 37 °C, 5% CO_2_ and 95% humidity for 48 h. The samples were rinsed with PBS, fixed with 4% formalin and blocked with 1% ABS tween. The first antibody, Integrin β1 (A-4): sc-374429 was added (50 µL per plate diluted with PBS-tween 1:150) and incubated at 4 °C overnight. After washing with PBS, the second antibody (normal mouse IgG: sc-3891 SCBT, was added (50 µL per plate diluted with PBS-tween 1:500) and left for 2 h. The samples were observed in a Leica fluorescence microscope equipped with a digital camera, using the LAS AF software (Leica, Wetzlar, Germany). The same procedure was carried out for vinculin, where the antibody (7F9 sc-73614), was used. All antibodies were provided by Santa Cruz Biotechnology (Dallas, Texas, USA).

## 3. Results and Discussion

### 3.1. Composition and Structure of Crosslinked CHT

#### 3.1.1. FTIR Spectroscopy 

FTIR spectra of chitosan films dried at 25 °C and at 150 °C are shown in Figure 1a and b respectively. Non crosslinked chitosan (Figure 1a) showed the typical broad absorption between 3677 cm^−1^ and 3000 cm^−1^ (peaks at 3360 cm^−1^ and 3285 cm^−1^) corresponding to the O–H (from chitosan and water) and vibration of N–H in N–H_2_ groups. At 2920 cm^−1^ and 2850 cm^−1^ absorptions corresponding to stretching C–H in C–H_2_ (C–H from the ring, C–H_2_ from non-acetylated and acetylated chitosan and C–H_3_ from acetylated chitosan) were observed. While the absorption band at 1650 cm^−1^ was associated to C=O vibration in the amide I N–H (acetamide) group, the band at 1560 cm^−1^ was attributed to C–N and N–H_2_ bending in amide II region. At 1422 cm^−1^ bands related to CH_2_ bending were observed. 

At 1380 cm^−1^ an absorption which can be related to CH_2_ wagging appeared; and finally, at 1319 cm^−1^, the amide III appeared. Amide III is commonly used for calculating the degree of acetylation. Skeletal vibrations characteristic of the chitosan structure appears at 1150 cm^−1^ (C–O–C), 1080 cm^−1^ and 1030 cm^−1^ (pyranose ring). 

When CHT was crosslinked with GA, the intense band at 1650 cm^−1^ became composed of the amide I and the corresponding imine bond [36,37] (–C=N–typically found at 1655 cm^−1^), however amide II was of higher intensity than amide I absorption. The Fourier-transform infrared spectroscopy (FTIR) spectra of chitosan films crosslinked with PEGDE showed a higher intensity of amide I in comparison with amide II. This difference was even more marked as the amount of PEGDE increased. The amino groups, now at 1577 cm^−1^, are more reactive than hydroxyl groups, and can react with the epoxy ring leading a crosslinked structure while showing lower intensity at the highest concentration of PEGDE (see (Appendix A) for possible chemical reactions). Pristine PEGDE exhibited bands at 1348 cm^−1^ and 1456 cm^−1^ which were expected to appear by PEGDE crosslinking as reported by Song et al. [38]. However, either these absorptions were not detected, or chitosan peaks overlapped them. Likewise, although N–H reduction was expected, the introduction of new O–H groups masked the predicted effect. 

The infrared spectra of the cross-linked films dried at 150 °C are presented in Figure 1b. The band located at 1640 cm^−1^ (amide I) was reduced significantly, in comparison with the band at 1560 cm^−1^ (amide II), which shifts on drying. During GA crosslinking, amide II was smaller than amide I, and the absorption at 1380 cm^−1^ remained unchanged, but at a lower intensity as to the band at 1420 cm^−1^. In PEGDE-crosslinked CHT, a higher intensity of the amide I was observed again, in comparison with amide II. This indicates that either these bands are not a good reference to estimate the degree of crosslinking, or that the crosslinking mechanism is different for GA rather than for PEGDE crosslinking. However, the broad O–H and N–H bands located at 3360 cm^−1^ were of higher intensity compared to the vibration of C–H located at 2922 cm^−1^ in contrast to the observed behavior in CHT alone and GA-crosslinked CHT. Finally, absorptions at 1380 cm^−1^ and 1315 cm^−1^ were of higher intensity compared to the band at 1418 cm^−1^.

Kumar-Krishnan et al. [30] reported that during the heating from 30 °C to 110 °C of neutralized chitosan films, two relevant changes were observed: The reduction of the intensity and shifting of the band centered at 3360 cm^−1^ to 3445 cm^−1^, and the shifting of the amide I band from 1635 cm^−1^ to 1655 cm^−1^. Our study shows that that drying at 150 °C reduces O–H intensity (see Figure 1a,b for pristine CHT). This was confirmed by the increase in intensity of the 2922 cm^–1^ and 2850 cm^−1^ absorptions. Therefore, not only adsorbed water would be eliminated, but also it can be expected the breakdown of intra- and/or inter-chains hydrogen bonds between the water molecules and the NH or O–H groups of the chitosan molecule. The reduction in O–H and N–H intensity was also observed in crosslinked films, however, it was less notorious on PEGDE crosslinked CHT as it is more hydrophilic. 

Regarding, the amide I band, there was no apparent shift; nevertheless, it exhibited a significant reduction in intensity. This agrees with Zawadzki et al. [32], in which is observed not only a decrease in the intensity of the broad band centered at approximately 3370 cm^−1^, but also in the absorption of the C=O group of chitosan (i.e., amide band I). This outcome shows that interaction of adsorbed water molecules is diminished not only in the 3400–3200 cm^−1^ region, but also in the carbonyl region, which also implies that dehydration takes place not only at the surface level, but also in the bulk of chitosan. 

#### 3.1.2. Determination of Free Amino Groups in Chitosan Films

Table 1 shows the percentage of amino groups of uncrosslinked and crosslinked CHT films dried at 25 °C and 150 °C. From this table, it is evident that amino groups are reduced by both crosslinking and thermal treatment at 150 °C, i.e., from 72.5% at 25 °C to 54.9% at 150 °C for CHT alone and from 53.4% at 25 °C to 49.9% at 150 °C for highly PEGDE crosslinked CHT. From these results, it is also clear that PEGDE can be as effective as GA for crosslinking, especially at higher concentrations.

Thermal treatment at 150 °C yields an insoluble chitosan scaffold when placed in acetic acid. After thermal treatment at 150 °C it is possible that some chemical reactions involving amino groups lead to the crosslinking of the chitosan scaffold, for example oxidation of alcohols render aldehydes, which in turn reacted with the amino groups, leading to imine formation. This explain why the percentage of the amino groups quantified at 150 °C was very low. 

These results should be taken with caution, as it is possible that acetic acid was either not completely eliminated, not only from dissolution and film preparation, but also from purification of the raw product during manufacture; or because commercial chitosan employed was in partially protonated form. In addition, Balázs [39] mentions that high acid concentrations result in more distant equivalence points during titration. In our study, there was no IR evidence of acetate salts remaining; yet, in solid state FTIR, it is possible that traces of acetic acid cannot be detected.

The binding and growth of cells in chitosan substrates has been attributed to the cationic nature of chitosan amine groups. Data suggest that, as the level of deacetylation is raising, the positive charge density of the chitosan increases; consequently, there is an increase in the attraction of negatively-charged cells. Factors such as the degree of deacetylation and chemical modifications influence the hydrophilic properties of chitosan films and their interactions with the biological environment [40,41]. These effects will be further discussed in terms of biocompatibility.

#### 3.1.3. Thermogravimetric Analysis (TGA) 

Thermal stability can be evaluated as the onset temperature of the degradation step, the maximum of the decomposition temperature and the temperature corresponding to the 50% weight lost. In this study we use the maximum temperature and the 50% weight lost to compare the thermal stability.

Figure 2 shows TGA thermograms of uncrosslinked, GA crosslinked and PEGDE crosslinked chitosan films treated at 25 °C (Figure 2a,c) and 150 °C (Figure 2b,d). The degradation profile has two main stages; still, some GA crosslinked CHT showed a third stage of degradation as reported by Lopez et al. [42]. CHT films dried at 25 °C and 150 °C exhibited two stages of weight loss. The first weight loss was approximately at 15% (65 °C) and 9% (68 °C), respectively. This weight loss is attributed to the elimination of water molecules that are adsorbed on the surface of chitosan [43].

It is observed that the previously dried CHT at 150 °C shows a lower weight loss as a result of the drying process that removes part of the adsorbed water. However, the weight loss related to water above 100 °C is due to hydrogen bonds formation between functional groups in CHT molecules and water [44]. In the second stage, dry chitosan samples at 25 °C and 150 °C showed a 50% (297 °C) and 72% (300 °C) of weight loss, respectively; it can be attributed to the partial decomposition of the structure of CHT [35]. The total weight loss for dry CHT at 25 °C was 65% and for dry CHT at 150 °C it was 81%.

The greater weight loss for CHT dried at 150 °C, compared to the dried at 25 °C, could be due to the fact that the second stage of chitosan degradation began around 150 °C [44], so the drying process could be affected by the thermal stability of the material. Decomposition temperatures and weight losses of CHT samples crosslinked with glutaraldehyde and with different concentrations of PEGDE are summarized in Table 2.

In relation to CHT crosslinked with PEGDE dried at 25 °C, a decrease in weight loss corresponding to adsorbed water is observed in comparison with pure chitosan. However, at higher PEGDE concentrations, adsorbed water remains the same. This could be associated with crosslinking between chitosan and PEGDE, which would decrease the absorption of water in the material [45]. Additionally, a slightly higher thermal stability (Td2) of the samples crosslinked with PEGDE is observed, in contrast to the pure CHT and the CHT crosslinked with glutaraldehyde.

Likewise, more water was present in samples dried at 25 °C than in those treated at 150 °C. However, the mass loss in the second stage of degradation was similar. For CHT dried at low temperatures, the amount of retained water was higher in GA crosslinked CHT than in PEGDE crosslinked CHT, demonstrating its higher efficiency as a crosslinker for rendering a low water absorbing hydrogel. 

For pristine CHT, water is lost in the first stage; whereas, in the second stage, chitosan main chain degradation (depolymerization) and the elimination of volatile products, from pyranose ring destruction and deamination, is expected as reported by Zawadski et al. [32]. Nonetheless, when the crosslinking agent is present, other compounds should be expected.

Finally, for all CHT samples crosslinked with PEGDE and dried at 150 °C, it is generally observed that they exhibited better thermal stability than pure chitosan and chitosan crosslinked with PEGDE at 25 °C. This could indicate that the drying treatment at 150 °C promotes a better crosslinking between the chitosan and the PEGDE, which renders samples with higher thermal stability. In other words, CHT-PEGDE dried at 150 °C exhibited a smaller loss of mass (53–54%) at interval of 300–700 °C compared to CHT (72%). In the same way, all samples of CHT-PEGDE dried at 150 °C revealed a smaller loss of mass than CHT-PEGDE at 25 °C (58–70%).

Petrova et al. reported the lost solubility of their samples, attributed to the formation of ionic crosslinks upon dehydration [46]; we observed the same effect in our films after drying at 150 °C.

#### 3.1.4. X-Ray Diffraction (XRD)

Figure 3 shows the diffractograms of the samples of chitosan dried at 25 °C and 150 °C. The diffraction pattern of the neutralized CHT films dried at 25 °C (see Figure 3a) showed the two characteristic sharp peaks at 2θ = 9.54° and 2θ = 20.3°, which are typical of the hydrated conformation of chitosan. The peak at low angle is attributed to the crystalline phase I, and the peak of high angle is attributed to the crystalline phase II, which presents less hydrated and more rigid chains dispersed in an amorphous phase [47,48]. 

This highly ordered structure arises from the hydroxyl and amino groups which can form strong intermolecular and intramolecular hydrogen bonds. This structure was retained with a slight degree of crosslinking (CHT PEGDE1), but giving rise to additional peaks at 2θ = 5.6° and 2θ = 15° in more crosslinked samples (CHT PEGDE2 and CHT PEGDE3). Baroudi et al. attribute the 2θ = 15° to a hydrated chitosan polymorph crystal as a complex with water and acid [37].

When GA was used as the crosslinking agent, the amorphous halo became more evident (peak at 2θ = 19.7°) which indicates the loss of the crystalline structure. Figure 3b shows the diffraction pattern of CHT films dried at 150 °C. It is clear that the drying process changing the CHT structure renders it more amorphous, i.e., although the same peaks were observed, they exhibited a lower intensity. On the contrary, the Rivero et al. [49] findings showed that CHT thermally treated at 160 °C during 30 min caused a shift from 2θ = 11° to 2θ = 15°.

Regarding the slightly PEGDE crosslinked CHT, while the peak at 2θ = 20.1° tended to disappear, the peak at 2θ = 9.58° not only reduced but shifted to lower angles—this outcome suggests an increase in chain separation. Interestingly, CHT PEGDE2 showed multiple peaks (2θ = 9.5°, 15° and 20.4°) denoting a heterogeneous structure; contrary to CHT PEGDE3, a more crosslinked sample, which clearly showed only two reflections at 2θ = 9.68° and 2θ = 20.2°.

There are reports that suggested thermal treatments on chitosan can change its physical properties, affecting not only its appearance but also its aqueous solubility and rheology [50,51,52]. Our results clearly show that the crystallinity of the film decreases after, i.e., currently, amino groups available for hydrogen bonding can be used in the reaction with the aldehyde group (GA) or epoxy ring (PEGDE) for the formation of an amorphous CHT [53]. For a more accurate comparison, Table 3 summarizes the results obtained from the calculation of the crystallinity index (Cr*I*) of each of the films. The table shows how crystallinity decreased with the degree of crosslinking and with drying at 150 °C.

It has been reported that the anhydrous chitosan produced through exhaustive dehydration processes has an identical crystal structure to hydrated chitosan. This is, characterized by having an orthorhombic unit cell with a = 7.76 Å, b = 10.91 Å and c = 10.30 Å (for a type I crystal) as well as a = 4.4 Å, b = 10.0 Å and c = 10.3 Å (for type II crystal). However, instead of containing four chains of chitosan with eight water molecules, only two chains were involved [48].

It was expected that more water was present in samples with a lower crystalline index (Cr*I*), as the water can easily penetrate amorphous regions. This was not observed, as Td1 was 15% when the sample was dried at 25 °C, and only 9% when the sample was dried at 150 °C for the pristine chitosan. This can be explained by the used formula and the presence of multiple peaks in the diffractogram. 

### 3.2. Surface Properties of Crosslinked CHT

#### 3.2.1. Elemental Composition by EDX and XPS

Elemental composition, as shown in Table 4, revealed that in samples dried at 25 °C oxygen tends to increase in parallel with the increase in the amount of PEGDE during crosslinking of chitosan; this is so due to oxygen present in the PEGDE structure. In the case of nitrogen, a slight decrease was observed as films were crosslinked. CHT films treated at 150 °C experienced also a slight increase in superficial oxygen content with respect to samples dried at 25 °C, but remained constant during crosslinking with either PEGDE or GA. In contrast, superficial nitrogen tends to reduce with thermal treatment and crosslinking with PEGDE, but not with GA. This indicates that more oxygen than nitrogen is available at the PEGDE crosslinked CHT.

However, XPS showed lower content of oxygen and nitrogen on the CHT surfaces; still, it followed the same trend as noted above, i.e., oxygen content increases in parallel with the increase in PEGDE concentrations, but nitrogen content remained constant. 

#### 3.2.2. X-ray Photoelectron Spectroscopy (XPS)

The XPS survey spectra of chitosan films dried at 25 °C and 150 °C are shown in Figure 4. As expected, only C1s, N1s and O1s were detected. However, a close inspection of pristine CHT dried at 25 °C revealed at least four types of carbons (C–C (284.5 eV), C–O (286.4 eV), C=O and C–N (288.5 eV) which were better resolved during its drying at 150 °C. In addition, from the three possible types of oxygen, C–OH, C–O–C and C=O, only two were detected at 532 eV and 533 eV, respectively, which render a symmetric O1s peak after drying at high temperatures. Finally, nitrogen should exhibit at least two types (–NH_2_ and O=C–N–H), but only one peak at 399.5 eV was observed, and broadens during its drying at 150 °C.

GA crosslinked CHT exhibited three types of carbons at 25 °C, but they tend to disappear with drying at 150 °C, i.e., the C1s band broadens. GA crosslinked CHT showed also a symmetrical O1s peak for films dried at 25 °C and 150 °C. Finally, N1s become broader as thermal treatment increased; however, three peaks (399 eV, 400.5 eV and 402.5 eV) at low temperature drying were observed. This is a clear evidence of the crosslinking reaction.

In PEGDE crosslinked samples, three types of C1s were observed, but when PEGDE at medium and high concentration was used, C–O (286.4 eV) were clearer (more at.% contribution) in samples treated at 25 °C; nevertheless, it regained its original intensity when dried at 150 °C. PEGDE crosslinked CHT showed also a symmetrical O1s peak for films dried at 25 °C and 150 °C. For the case of N1s, no visible changes were observed at low concentrations (PEGDE1), at both drying conditions, but N1s became either broader or with two broad peaks (399 eV and 402.5 eV) at the higher concentration (PEGDE 3) when dried at 25 °C. Appendix A show the corresponding XPS scans for C, N and O.

#### 3.2.3. Surface Topography by SEM and AFM

Figure 5 shows the topography of films dried at different conditions. A change in coloration was the only visible signal as films changed from light yellow to brown/red with the thermal treatment at 150 °C. However, SEM did not detect changes in the surface morphology of the films with the additional drying treatment at 150 °C, in contrast with Lewandowska et al. who reported a well-defined island structure which can result from the crystallinity of the sample [54].

Atomic force microscopy was also used to evaluate the surface topography of the films. When comparing the mean surface roughness (*R*a) of the films dried at 25 °C (Table 5), it was observed that pristine CHT and PEGDE crosslinked CHT, at the lowest concentration (PEGDE1), showed low Ra values (44–45 nm) which increased with PEGDE concentration (64.1 nm and 91.2 nm for PEGDE 2 and PEGDE3 respectively). Surprisingly, the smoothest surface was achieved with GA crosslinked CHT as the *R*a values was 26.3 ± 6.3 nm. As the films were dried at 150 °C, the sample that showed highest *R*a was GA crosslinked CHT, uncrosslinked CHT with a *R*a = 103.7 ± 18.6nm, and *R*a = 75.9 ± 12.1 nm, respectively. In contrast, PEGDE crosslinked CHT showed a decrease in *R*a compared to those dried at 25 °C i.e., 28.2 ± 7.9 nm, 44.7 ± 10.8 nm and 16.5 ± 2.3 nm for PEGDE1, PEGDE2 and PEGDE3, respectively. Factors such as roughness, porosity and surface topography can influence the cellular behavior of a material [41]. In this regard, Croisier et al. [18] reported a decrease in roughness on chemically-modified surfaces with PEG showing greater uniformity. They suggest that the PEG seemed to soften the surface topography, which confirms the presence of the PEG, which was chemisorbed on the surface, being the films with PEG smoother and hydrophilic [16]. In our study, this effect was only observed on films with the additional drying of 150 °C. See Appendix A.

#### 3.2.4. Measurement of Contact Angle

Based on the measurement of contact angles in water, Drelich et al. [47] propose the following classification: Hydrophilic surfaces are those in which the water extends completely, visually “zero contact angle” (0°); weakly hydrophilic (56–65°) and weakly hydrophobic (>56–65°) are those in which water films are unstable; and, hydrophobic surfaces are those with contact angles of at least 90° [55].

The hydrophilicity of the surfaces of the crosslinked CHT films with the different drying treatments (25 °C and 150 °C) are shown in Table 5. CHT films showed, in general, water contact angles lower than 72°, which classified them as weakly hydrophilic [47]. However, thermal treatment at 150 °C slightly increased the contact angle for pristine CHT, and PEGDE highly crosslinked samples and had no effect on GA crosslinked samples. When DMEM was used as the sessile drop, lower contact angles were observed in crosslinked samples, being the lowest with PEGDE and GA.

Polar groups introduced by different procedures tend to be buried in bulk when they enter in contact with air for a prolonged period; however, they remain on the surface when they are in contact with water or any other polar environment [55]. In our case, hydrogen bond acceptor groups (namely O atoms) either as ethylene glycol units or as O–H produced during epoxy ring opening increased the water affinity of the amino-PEGDE surface [56].

These results agree with oxygen surface composition obtained by EDX, as it was clear that oxygen-containing groups were at the surface rendering a more hydrophilic surface. Zhang et al. [49] report similar results with PEG, where the contact angle was reduced somewhat by adding PEG but did not change much or even slightly increased when the proportion of chitosan to PEG was increased. In contrast, these results were unexpected since a higher Ra value should have led to a lower contact angle. Therefore, it seems that the surface chemical composition plays a more important role in determining contact angle behavior.

In general, surface wettability is strongly associated, not only with surface contact angle, but also with surface free energy, and it is affected by diverse aspects such as surface elemental composition and roughness. Wettability is perhaps the most important factor to determine the quantity and quality of adsorbed proteins, which in turn affects cell adhesion [57]. 

Yinghui et al. [41] noted that the high biocompatibility of chitosan is due to its high hydrophilicity and positive surface charge which promotes cell growth. Moreover, some studies have indicated that smaller contact angles in materials can increase affinity [58,59]. However, researchers have shown that instead of a direct correlation between the contact angle and the cell junction, there seems to be a contact angle ideal that best directs proliferation and cellular behavior, and this occurs around 60°–80° [29,60]. Still, there are clear data from many authors that indicate that this is not necessarily true, and that the contact angle is not a good predictor of cell union and behavior [29].

### 3.3. Cell Viability and Proliferation Studies 

Chitosan scaffolds must be able to support cell adhesion, and the growth and development of the extracellular matrix, as well. Cell adhesion and proliferation in chitosan substrates has been attributed to the cationic nature of chitosan amine groups [40].

Figure 6a shows the cell viability in direct contact (D) and indirect contact (I) with uncrosslinked and crosslinked CHT scaffolds dried at 25 °C. Also, this figure shows that osteoblasts in contact with extracts render similar or higher cell viability than when they are tested in direct contact (the equivalent to a more concentrated extract). It was also clear that the highest percentage of cell viability was observed for PEGDE crosslinked CHT at the intermediate (PEGDE2) and minimum (PEGDE1) concentration. Among all the samples studied, those crosslinked with GA exhibited the lowest cell viability.

Figure 6b shows the corresponding cell viability for scaffolds treated at 150 °C, in which a greater cellular viability was also observed for PEGDE1 and PEGDE2 crosslinked CHT. However, cell viability in direct contact also decreased on scaffolds dried at 150 °C for pristine CHT (lower absorbance). 

This means that high cell viability in direct contact was achieved in pristine semicrystalline (CrI = 64.1%) CHT dried at 25 °C with 72.5% of free amino groups, Ra = 45.8 ± 4.6 and 72° of DMEM contact angle and in semicrystalline (CrI = 51.3%) PEGDE1 crosslinked CHT dried at 150 °C with 55.3% of free amino groups, Ra = 28.2 ± 7.9 and 57° of DMEM contact angle. This also means that, for a lower amount of amino groups, a higher hydrophilicity renders higher viability in the presence of a less semicrystalline structure. GA crosslinked CHT dried at 25 °C exhibited the lowest viability in direct contact with 51.5% of free amino groups, Ra = 26.3 ± 6.3 and a DMEM contact angle of 56°. Even when the contact angle is lower than pristine CHT, less amino groups are available, resulting in a lower viability. Furthermore, glutaraldehyde is considered as a toxic crosslinking agent that rendered an amorphous (CrI = 21.5%) and smooth CHT surface allowing little cell attachment.

#### 3.3.1. Analysis of Cellular Adhesion by Scanning Electron Microscopy (SEM)

The distribution of the cells in scaffolds dried at 25 °C and 150 °C was observed after 48 h of incubation by either direct or indirect contact (Figure 7). A better interaction and osteoblast morphology are observed in direct contact in PEGDE crosslinked CHT thermally treated at 150 °C. The presence of rounded osteoblast in GA crosslinked CHT instead of polygonal cells suggests a response to a stimulus, but not in viability.

Jayakumar et al. [61] evaluated the cytotoxicity of chitosan/polyethylene glycol diacrylate (PEGDA) scaffolds with fibroblasts (L929), and reported that the material showed no cytotoxicity towards cell growth and had a good compatibility in vitro. The SEM observation indicated that the microporous surface structure of the chitosan/PEGDA scaffolds was good for growing, proliferating and differentiating cells [61].

#### 3.3.2. Vinculin and Actin Expression Analysis 

Vinculin is a cytoskeletal protein associated with cell-cell and cell-matrix junctions, and also to cell-biomaterial interactions. Figure 8 shows the expression of vinculin for crosslinked CHT with drying treatments at 25 °C and 150 °C. It is clear that lightly PEGDE crosslinked CHT (PEGDE1 and PEGDE2) dried at 150 °C exhibited strong protein expression in multiple elongated cells, even better than in CHT alone. This implies that contractile forces are enhanced due to the vinculin presence. On the contrary, scaffolds crosslinked with GA showed irregular cell shape. 

The ability to generate and transmit forces depends on the strength of the connection between the integrin adhesion receptors and the actomyosin cytoskeleton mediated by focal adhesion proteins. Vinculin connects the actomyosin cytoskeleton with β integrin subunits through binding to talin, or with the α4 and α9 integrin subunits through binding to paxillin, respectively. In the adhesion test, using the integrin antibody the obtained images are shown in Figure 8. It was observed a greater number of cells in PEGDE crosslinked CHT scaffolds compared to CHT without crosslinker and CHT-GA. These findings indicate that PEGDE improves cell adhesion. A better cell morphology was observed in the scaffolds with additional drying of 150 °C. This was more evident in the PEGDE cross-linked CHT, when the crosslinker was used at the lowest concentration

The characteristics of the polymer surface could affect the amounts and types of bound proteins, as well as the conformation, orientation, or binding strength of the adsorbed protein. Zhang et al. [49] reports that hydroxyl groups and hydrophilic PEG chains significantly improved protein adsorption and cell growth. Chitosan is an unusual polysaccharide with a positive charge which promotes cell growth and protein adsorption, unlike most proteins which have isoelectric points below 7.4 and a negative charge. The introduction of the appropriate amount of PEG may not influence this property, but high concentrations of PEG may reduce the positive charge [57].

Although all PEGDE crosslinked CHT scaffolds support cell adherence, it is worth bearing in mind that that cell adhesion, as demonstrated in this study, is not indicative of how supportive a substrate is to cell spreading, and that cell spreading does not correlate with focal contact formation [62].

## 4. Conclusions

The results obtained demonstrate that there are several chemical and structural differences between CHT scaffolds dried at 25 °C and 150 °C, as shown by various techniques. FTIR showed that amide II/amide I ratio changed as the drying temperature was increased in pristine CHT and GA crosslinked CHT, and that the formation of a more amorphous polymer or with reduced crystallinity was observed as demonstrated by XRD. 

In addition, the decomposition temperature (Td2) tends to increase as demonstrated by TGA. When the scaffolds were crosslinked with PEGDE at medium and high concentrations, a similar behavior was observed, i.e., a reduction in free amino groups, amine I/II ratio and crystallinity. 

However, osteoblast viability and adhesion was promoted by PEGDE crosslinked CHT at low and medium concentrations, mainly due to its hydrophilicity (low contact angles due to the presence of oxygen containing species), its low crystallinity and its low surface roughness, in spite of the small reduction in the amount of free amino groups on the surface induced during drying at 150 °C. 

Finally, these results suggest that PEGDE at low concentrations is a suitable biocompatible crosslinker agent for chitosan; and, consequently, these materials are can be used in bone contact applications

## Figures and Tables

**Figure 1 polymers-11-01830-f001:**
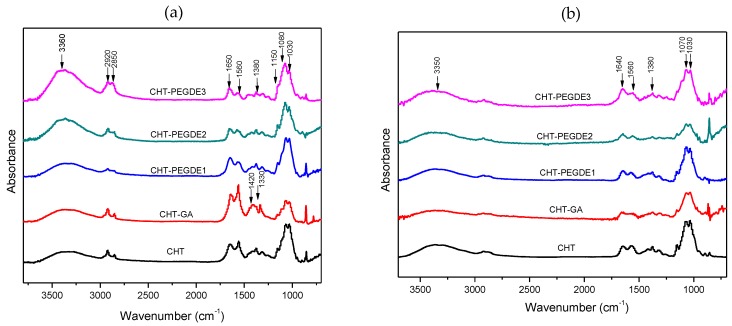
(**a**) Fourier-transform infrared spectroscopy (FTIR) spectra of Chitosan (CHT) films dried at 25 °C (**b**) FTIR spectra of CHT films dried at 150 °C.

**Figure 2 polymers-11-01830-f002:**
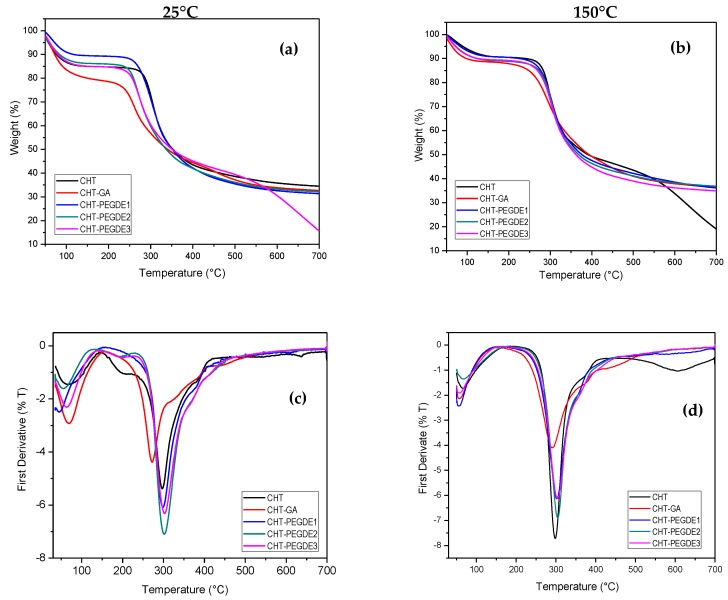
Thermogravimetric Analysis (TGA) (**a**,**b**) and DTGA (**c**,**d**) thermograms of Chitosan (CHT), glutaraldehyde crosslinked chitosan (CHT-GA) and polyethylene glycol diglycidyl ether (PEGDE) crosslinked chitosan.

**Figure 3 polymers-11-01830-f003:**
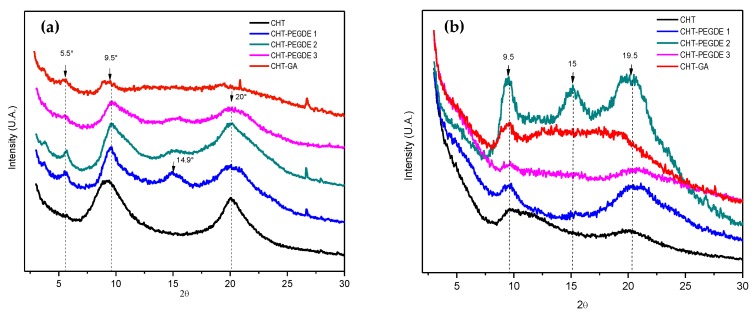
XRD pattern of CHT, glutaraldehyde crosslinked CHT (CHT-GA), PEDGE crosslinked CHT (CHT-PEDGE) at 0.114 mM (1), 0.228 mM (2) and 0.342 mM (3), dried at 25 °C (**a**) and 150 °C (**b**).

**Figure 4 polymers-11-01830-f004:**
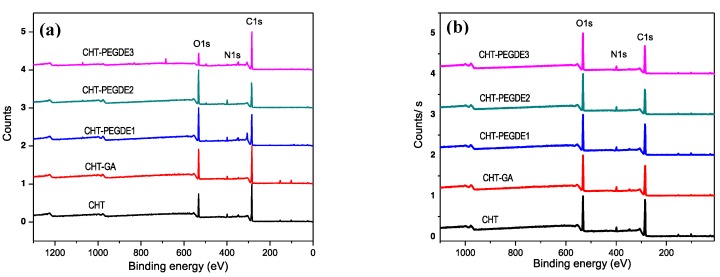
XPS survey spectra of CHT, glutaraldehyde crosslinked CHT (CHT-GA), PEDGE crosslinked CHT (CHT-PEDGE) at 0.114 mM (1), 0.228 mM (2) and 0.342 mM (3) dried at 25 °C (**a**) and dried at 150 °C (**b**).

**Figure 5 polymers-11-01830-f005:**
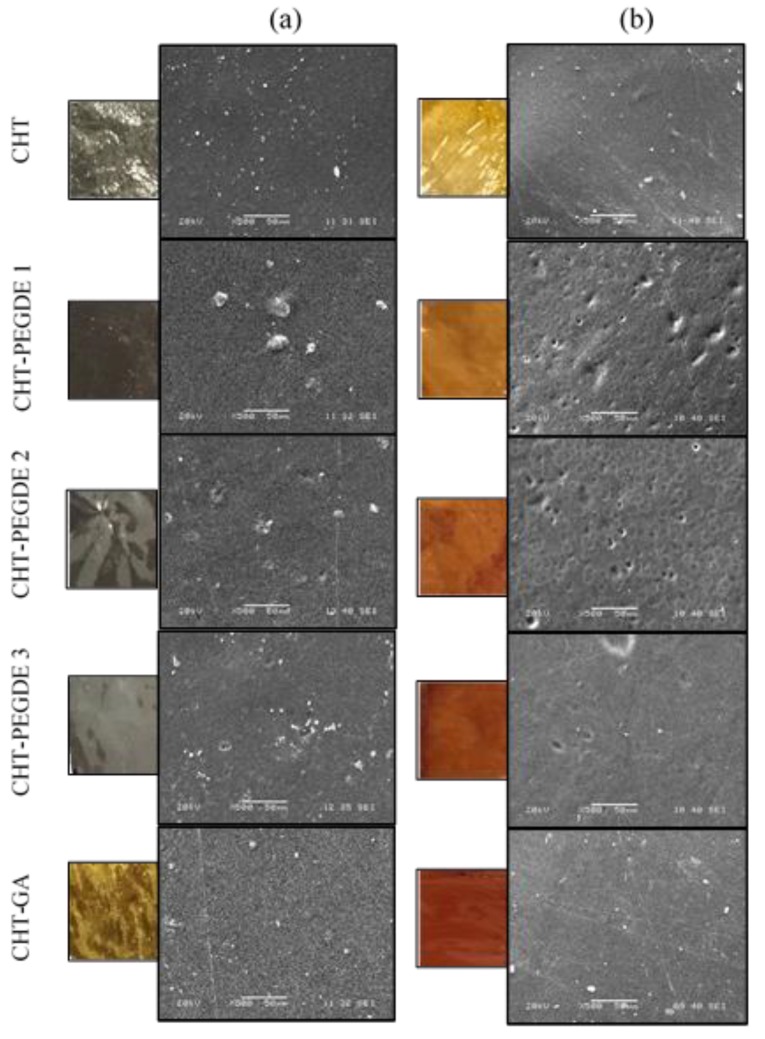
SEM surface morphology of CHT films dried at 25 °C (**a**) and 150 °C (**b**).

**Figure 6 polymers-11-01830-f006:**
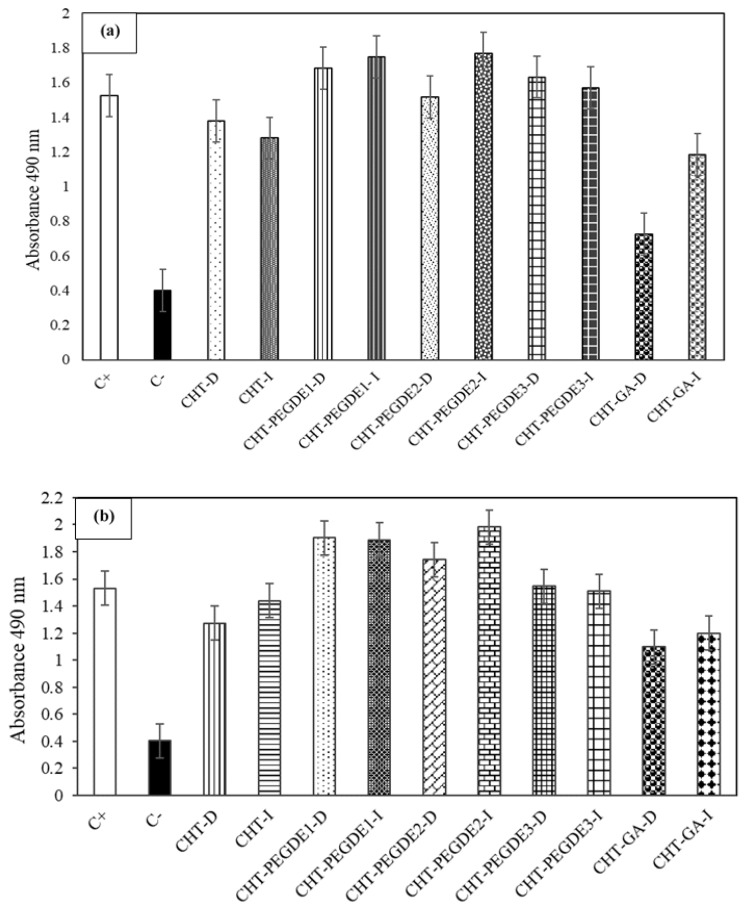
(**a**) Viability of osteoblast on direct (D) and indirect (I) contact with Chitosan (CHT), glutaraldehyde crosslinked chitosan (CHT-GA), PEDGE crosslinked chitosan (CHT-PEDGE) at 0.114 mM (1), 0.228 mM (2) y 0.342 mM (3) dried at 25 °C and (**b**) dried at 150 °C.

**Figure 7 polymers-11-01830-f007:**
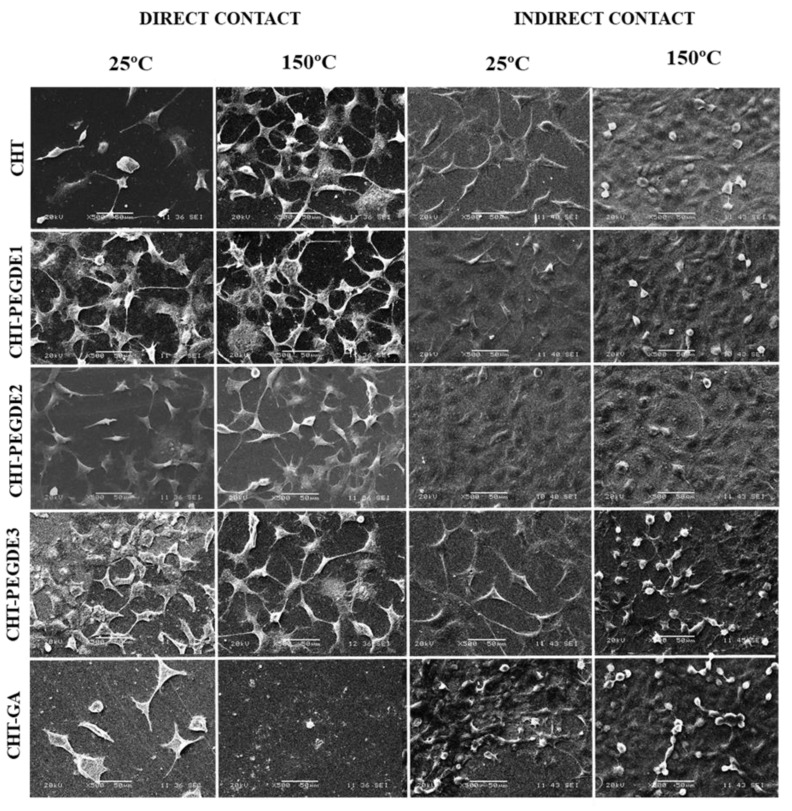
SEM images of osteoblast seeded onto CHT scaffolds on direct and indirect contact dried at 25 °C and direct and indirect contact dried at 150 °C.

**Figure 8 polymers-11-01830-f008:**
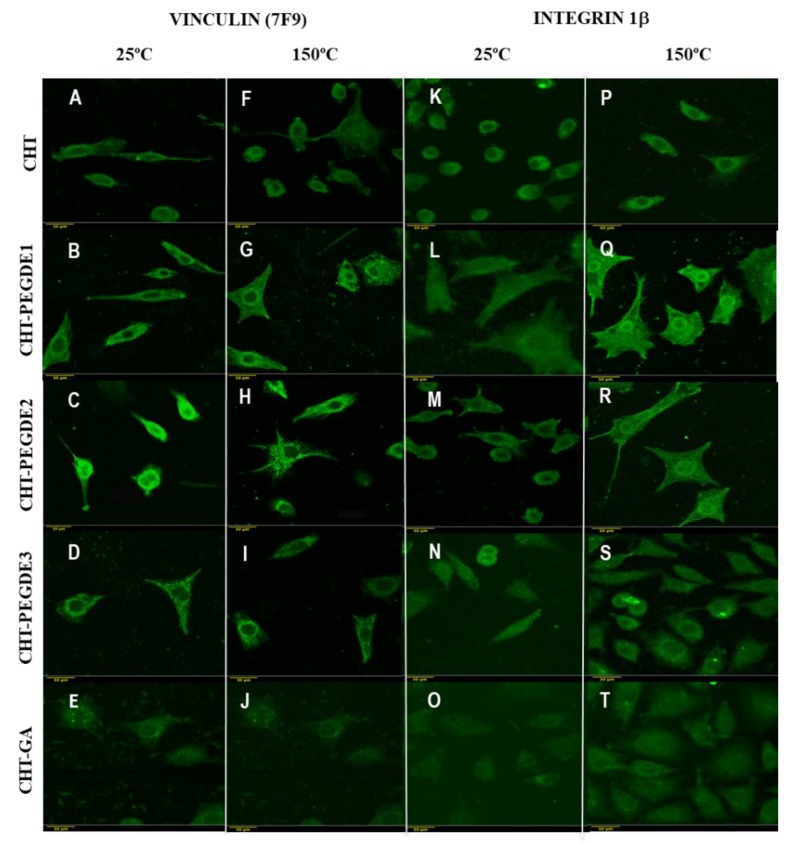
(**A**–**J**) Fluorescence images of osteoblast seeded onto CHT scaffolds labeled with vinculin (7F9): sc-73614, cultivated for 48 h. (**K**–**T**) Fluorescence images of osteoblast seeded onto CHT scaffolds labeled with intregrin 1β (A-4), cultivated for 48 h.

**Table 1 polymers-11-01830-t001:** Percentage of amine groups in chitosan and crosslinked chitosan films, dried at 25 °C and with additional drying at 150 °C.

	Amine Groups (%)	
	25 °C	150 °C
**CHT film**	72.5 ± 3.8	54.9 ± 7.1
**CHT-PEGDE1**	56.5 ± 4.1	55.3 ± 3
**CHT-PEGDE2**	55.4 ± 5.6	52.3 ± 2.2
**CHT-PEGDE3**	53.4 ± 4.3	49.9 ± 5.2
**CHT-GA**	51.5 ± 2.5	47.7 ± 1.9

**Table 2 polymers-11-01830-t002:** Maximum temperatures and weight loss of CHT films dried at 25 °C and 150 °C.

Films	*T*d (°C)				% Weight Loss				*T* (°C) at 50% of Weight Loss	
	**25 °C**		**150 °C**		**25 °C**		**150 °C**		**25 °C**	**150 °C**
	*Td1*	*Td2*	*Td1*	*Td2*	*Td1*	*Td2*	*Td1*	*Td2*		
CHT	65	297	68	300	15	50	9	72	351	390
CHT-PEGDE 1	45	298	66	303	10	58	10	54	350	376
CHT-PEGDE 2	55	302	59	304	14	54	11	53	337	368
CHT-PEGDE 3	64	302	54	304	15	70	10	54	347	361
CHT-GA	68	272	55	292	21	47	12	51	333	392

**Table 3 polymers-11-01830-t003:** Percentage of the crystallinity of the chitosan films without crosslinker and with crosslinker of GA and PEGDE dried at 25 °C and 150 °C.

	Cr*I*%	
Films	25 °C	150 °C
CHT	64.17	40.33
CHT-PEGDE 1	64.17	51.33
CHT-PEGDE 2	37.39	7.64
CHT-PEGDE 3	52.58	26.79
CHT-GA	21.59	8.36

**Table 4 polymers-11-01830-t004:** XPS and EDX atomic percentage of C, O, N of chitosan films crosslinked with GA and PEGDE dried at 25 °C and 150 °C.

	CHT				CHT-PEGDE1				CHT-PEGDE2				CHT-PEGDE3				CHT-GA			
	25 °C		150 °C		25 °C		150 °C		25 °C		150 °C		25 °C		150 °C		25 °C		150 °C	
%	*XPS*	*EDX*	*XPS*	*EDX*	*XPS*	*EDX*	*XPS*	*EDX*	*XPS*	*EDX*	*XPS*	*EDX*	*XPS*	*EDX*	*XPS*	*EDX*	*XPS*	*EDX*	*XPS*	*EDX*
**C**	70 ± 1	55 ± 4	70 ± 4	53 ± 5	72 ± 2	52 ± 2	68 ± 4	58 ± 7	70 ± 4	53 ± 1	69 ± 2	59 ± 4	67 ± 3	50 ± 1	70 ± 6	60 ± 6	68 ± 3	57 ± 1	72 ± 4	56 ± 7
**O**	25 ± 2	37 ± 0	26 ± 4	41 ± 5	24 ± 3	41 ± 1	28 ± 3	40 ± 6	26 ± 4	41 ± 1	27 ± 4	40 ± 5	29 ± 2	44 ± 1	26 ± 4	39 ± 5	28 ± 3	37 ± 1	24 ± 2	39 ± 4
**N**	5 ± 1	8 ± 1	4 ± 0	6 ± 1	4 ± 0	7 ± 1	4 ± 0	2 ± 1	4 ± 0	6 ± 0	4 ± 1	1 ± 0	4 ± 1	6 ± 0	4 ± 0	1 ± 0	4 ± 1	6 ± 0	4 ± 0	5 ± 1

**Table 5 polymers-11-01830-t005:** Roughness and contact angle of the films dried at 25 °C and additionally dried at 150 °C.

	*R*a (nm)		Contact Angle (°)			
Film	25 °C	150 °C	25 °C		150 °C	
			H_2_O	DMEM	H_2_O	DMEM
**CHT**	45.8 ± 5	75.9 ± 121	62 9 ± 3	72 ± 4	71 ± 5	66 ± 6
**CHT-PEGDE1**	44.3 ± 4	28.2 ± 8	66 ± 2	65 ± 4	63 ±	57 ± 5
**CHT-PEGDE2**	64.1 ± 7	44.7 ± 11	70 ± 4	60 ± 4	72 ± 4	45 ± 4
**CHT-PEGDE3**	91.2 ± 5	16.5 ± 2	63 ± 7	56 ± 5	66 ± 4	62 ± 3
**CHT-GA**	26.3 ± 6	103.7 ± 19	68 ± 2	56 ± 2	68 ± 2	58 ± 7

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
