# Peer review of "The Effect of PEGDE Concentration and Temperature on Physicochemical and Biological Properties of Chitosan"

_polymers, 2019, doi:10.3390/polym11111830_

Round 1
Reviewer 1 Report
The Effect of PEGDE Concentration and Temperature on Physicochemical and Biological Properties of Chitosan manuscript is clear, concise and easy to read. The topic is within the scope of the journal and it is of interest for the readers.
There are a few minor changes that could improve the quality of the manuscript as for example:
Scaffolds should be mention in the abstract and keywords Line 59-60: The sentence should be re-written because should be removed Line 62mention the structure changes Methods: stirring method and conditions as G factor should be mention. Line 185-188: The description of the method should be further explain. Which were the manufacture instructions? The materials should be move to materials section. Tables should include the standard deviation of the data. Line 367 and 370: Likewise is repeated Line 644 the conclusion should be improved explaining the the CHT structure influence on.... due to the changes on....
Reviewer 2 Report
General comments/questions:
Lines 61-68: Give some literature review about current knowledge in factors affecting CHT biodegradation Was DDA provided by CHT producer or determined by Authors? (line 73) Explain: “0.016 is the NH2 content in 122 g/mL of 1M HCl” and specify what “theoretical NH2 content in CHT” mean In contact angle studies glycerol, diiodomethane and water are commonly used. Please provide characteristic of DMEM – is this a hydrophilic or hydrophobic liquid? In reviewer opinion FTIR part should be carefully read. All phrases like “NH groups” or “CH groups” should be corrected as particular bands in the FTIR spectra corresponds to “vibration of N-H in NH2 groups” Possible interactions “crosslinks” between CHT and both crosslinkers should be shown. Explain, why there are only such a small differences in NH2 content after crosslinking for membranes treated with higher T. Provide parameter used for comparison of “thermal stability” (see line 380). Generally, the onset temperature of degradation step should be analyzed or temperature corresponding to 50% weight loss. Provide specific values. Compare TGA analysis referring to crystallinity index. It is well know that water can easily penetrate only amorphous regions thus CrI should strongly affect hydration.
Particular comments:
Style: lines 27 (increasing concentration?), lines 90-91, Provide reference or partial definition of “biodegradable metals” (line 43) Line 45: give some examples of these unexpected properties Correct definition of chitosan (line 46): CHT is a copolymer composed of two types of units (with DDA higher than 50-60%). When DDA is < 60% then we can regard it as a chitin. Explain statement “superhydrophobicity [12],” – it is well known that chitosan possesses NH2 and OH hydrophilic groups and swell in water Line 62: add some examples of such structural changes Line 73: remove all decimal places in Mw Check carefully all indexes as in many parts typing error like “H20” were noticed In 2.3.5 – it should be noticed that EDX studies were performed for uncoated samples (especially in the first sentence coating was described) Lines 364-365: sentence unclear
